# Characterization of gravity waves in the lower ionosphere using VLF observations at Comandante Ferraz Brazilian Antarctic Station

Emilia Correia[1,2], Luis Tiago Medeiros Raunheitte[2], José Valentin Bageston[3], Dino Enrico D´Amico[2]

[1]Instituto Nacional de Pesquisas Espaciais, INPE, São José dos Campos-SP, Brazil
[2]Centro de Rádio Astronomia e Astrofísica Mackenzie, Universidade Presbiteriana Mackenzie, São Paulo-SP, Brazil
[3]Centro Regional Sul de Pesquisas Espaciais, CRS/INPE, Santa Maria-RS, Brazil

*Correspondence to*: Emilia Correia (ecorreia@craam.mackenzie.br)

**Abstract.** The goal of this work is to investigate the gravity waves (GWs) characteristics in the low ionosphere using very low frequency (VLF) radio signals. The spatial modulations produced by the GWs affect the conditions of the electron density at reflection height of the VLF signals, which produce fluctuations of the electrical conductivity in the D-region that can be detected as variations in the amplitude and phase of VLF narrowband signals. The analysis considered the VLF signal transmitted from the US Cutler/Marine (NAA) station that was received at Comandante Ferraz Brazilian Antarctic Station (EACF, 62.1$^o$ S, 58.4$^o$ W), with its great circle path crossing longitudinally the Drake Passage. The wave periods of the GWs detected in the low ionosphere are obtained using the wavelet analysis applied to the VLF amplitude. Here the VLF technique was used as a new aspect for monitoring GW activity. It was validated comparing the wave period and duration properties of one GW event observed simultaneously with a co-located airglow all-sky imager both operating at EACF. The statistical analysis of the seasonal variation of the wave periods detected using VLF technique for 2007 showed that the GW events occurred all observed days, with the waves with period between 5 and 10 min dominating during night hours from May to September, while during daytime hours the waves with period between 0 and 5 min are predominant all over the year and dominate all days from November to April. These results show that VLF technique is a powerful tool to obtain the wave period and duration of GW events in the low ionosphere, with the advantage to be independent of sky conditions, and can be used during all day and year-round.

Keywords: gravity waves, ionosphere, VLF, wavelet, Antarctica

## 1 Introduction

The upper part of the middle atmosphere, the upper Mesosphere and Lower Thermosphere (MLT), is dominated by the effects of the atmospheric waves (acoustic-gravity waves, gravity waves, tides and planetary waves) with periods from few seconds to hours, which are originated at tropospheric and stratospheric layers or even in situ generation. The waves with period below the acoustic cut-off, which is typically less than few minutes, are classified as acoustic wave, and the waves with period above the Brunt-Vaisala period, which is typically about 5 min, are classified as gravity wave (Beer, 1974).

During last decades, due to the recognized importance of the gravity waves (GW) in the general circulation, structure and variability in the MLT, and as an essential component in the Earth climate system (Fritts and Alexander, 2003; Alexander et al., 2010), these waves had been intensively investigated. For example, Earn et al. (2011) using data from SABER instrument onboard TIMED satellite, estimated the horizontal gravity wave momentum flux and showed that the fluxes at stratospheric heights (40 km) are stronger at latitudes above 50$^o$ in local winter and near the subtropics in the summer hemisphere, which are in agreement with Wang et al. (2005) and Zhang et al. (2012) that used temperature soundings of the same instrument and shows high gravity wave activity over regions of strong convection located at lower latitudes in summer and over the southern Andes and Antarctica Peninsula in winter. The sources of mesospheric GW obtained through high-resolution general circulation model also shows that the dominant sources are steep mountains and strong upper-tropospheric westerly jets in winter and intense subtropical monsoon convection in summer (Sato et al., 2009). Thus, any major disturbances that occur in the stratosphere can significantly modify the GW fluxes, which in turns change

the thermal and winds structures of the MLT region. One of these disturbances are the sudden stratospheric warmings (e.g. Schoeberl, 1978), which are large-scale perturbation of the polar winter stratosphere where the gradients of winds and temperatures are reversed for periods of days to weeks.

Acoustic-gravity waves (AGWs) and GWs are generated simultaneously by the same tropospheric sources and produce strong temperature perturbations in the thermosphere (e.g., Snively, 2013). The atmospheric gravity waves are originated in the lower atmosphere and propagate upwards, travelling through regions with decreasing density, which results in an exponential grow of their amplitudes (e.g. Andrews et al. 1987). The large wave amplitudes lead to wave breaking that deposits the momentum flux at the MLT region, which comes mostly from waves with periods lower than 30 min (Fritts and Vincent, 1987; Vincent, 2015). Theoretical, numerical and observational studies have improving the understanding of the GW sources, observed parameters (wavelength, period and velocity), propagation directions (isotropic/anisotropic), spectrum of intrinsic wavelengths and periods, and moment fluxes, as well as their impact in the MLT region. A variety of techniques has been used to obtain wave parameters, such as the horizontal and vertical wavelengths, phase speeds and periods, involving satellite observations as well as ground-based instrumentation. Each technique has its own strengths and limitations as presented, for example, by Vincent (2015).

The GW activity has been extensively observed mainly by using airglow all-sky imagers that permit to obtain the horizontal wave parameters and the propagation directions of the small-scale waves (e. g. Taylor et al., 1995). In airglow imagers the GWs are seen as intensity variations of the optical emission from airglow layers located at the MLT region (80 - 100 km altitude), but this technique requires dark and cloud-free conditions during the night, and particularly at high latitudes it is impossible to observe the nightglow during the summer since there are no totally dark condition during this season.

In order to avoid the limitations of the optical airglow observations, other techniques using radio soundings started to be used to characterize the mesospheric GWs in the ionospheric D- and E-regions. The propagation of GWs through the mesosphere induces spatial modulations in the neutral density, which modulates the electron production rate and the effective collision frequency between the neutrals components and electrons in the lower ionosphere. The ionospheric absorption of the cosmic radio noise is a function of the product of these two parameters, and so the fluctuations produced by the effect of GWs can be detected by imaging riometers. The ionospheric absorption modulations observed with different riometer beams permit to infer the gravity wave parameters such as the phase velocity, period and direction of propagation, as demonstrated by Jarvis et al. (2003) and Moffat-Griffin et al. (2008). They validated this technique comparing mesospheric GW signatures observed by using both co-located imaging riometer and airglow imager. AGWs in the ionosphere have been mapped using Global Positioning System total electron content data, and as reported by Nishioka et al. (2013), both AWs and GWs, are often observed to persist over hours.

The atmospheric gravity waves also can be detected in lower ionosphere using very low frequency (VLF: 3-30 kHz) radio signals. The amplitude and phase of VLF signals propagating in the Earth-ionosphere waveguide are affected by the conditions of the local electron density at reflection height, which is in the ionospheric D-region. The spatial modulations produced by the GWs in the neutral density produce

fluctuations of the electrical conductivity in the D-region, which are detected as variations in the amplitude and phase of VLF narrowband (NB) signals. AGWs have been detected as amplitude variations of VLF signals associated with solar terminator motions (Nina and Cadez, 2013), with the passage of tropical cyclones crossing the transmitter-receiver VLF propagation path (Rozhnoi et al., 2014), and particularly during nighttime, in association with local convective and lightning activity (Marshall and Snively, 2014). Planetary wave signatures also have been detected in the VLF NB amplitude data, which effects are pronounced during wintertime and present a predominant quasi 16-day oscillation (Correia et al., 2011, 2013; Schmitter, 2012; Pal et al., 2015).

The advantage of usage radio techniques to observe AGWs instead of the optical ones is that they are able to provide observations independently of the sky conditions, even during the daytime, and year-round. The purpose of this paper is to presents the characterization of the GW events detected in the lower ionosphere from the analysis of the VLF NB amplitude of signals detected at Comandante Ferraz Brazilian Antarctic Station (EACF). The wave parameters such as the period and the time duration of the GW activity will be obtained from the spectral analysis of the VLF amplitude fluctuations. The methodology using VLF technique is validated comparing the derived parameters of one GW event detected simultaneously with a co-located airglow all-sky imager.

## 2 Instrumentation and data analysis

The VLF signals propagate over long distances via multiple reflections, with considerably low attenuation, and are detected by VLF receivers after being reflected in the lower ionosphere at ~70-90 km of height (e.g., Wait and Spies, 1964). The changes detected in the amplitude and phase of the VLF NB signals give information of the D-region physical and dynamic conditions along the transmitter-receiver Great Circle Path (GCP), which are associated with the ionosphere electrical conductivity. This analysis uses VLF signals transmitted from the US Navy stations at Cutler/Maine (24.0 kHz, NAA) and at Lualualei/Hawaii (21.4 kHz, NPM), which after propagating along the GCPs NAA-EACF and NPM-EACF were detected with 1 sec time resolution using a AWESOME receiver (Scherrer et al. 2008) operating at EACF ($62.1^{o}$ S, $58.4^{o}$ W) station located on King George Island in the Antarctic Peninsula (Fig. 1).

The GW parameters were obtained from the VLF NB amplitude signals using a wavelet spectral analysis, which gives the wave period and time duration of GW activity, as will be described in the following section. To demonstrate the potentiality of usage of the VLF technique to observe GWs, the spectral analysis is applied during the night of June 10, 2007, when a prominent GW event (mesospheric front) occurred and it was well observed and characterized by using a co-located airglow imager along with temperature profiles form TIMED/SABER and horizontal winds from a medium frequency (MF) radar operated at Rothera station (Bageston et al., 2011). Afterwards, a year-round climatology of GWs of parameters related to the wave periods was obtained from the amplitude data of VLF signals propagating in the NAA-EACF GCP for the full year of 2007.

## 2.1 Wavelet spectral analysis

The Wavelet analysis was used to obtain the parameters of VLF amplitude signal fluctuations, which might be associated with the time and duration of the GW event and the period range it covers. The tool used is developed by Torrence and Compo (1998) and includes the rectification of the bias in favor of large scales in the wavelet power spectrum, which was introduced by Liu et al. (2007). The analysis uses the Morlet mother wavelet with frequency parameter equal 6, significance level of 95 % and time lag of 0.72 (Torrence and Compo, 1998). The wavelet analysis returns the following general results: the Power Spectrum; the Global Wavelet Spectra, which measures the time-averaged wavelet power spectra over a certain period and its significance level; and scale-averaged wavelet power, that is the weighted sum of the wavelet power spectrum over 2 to 64 band.

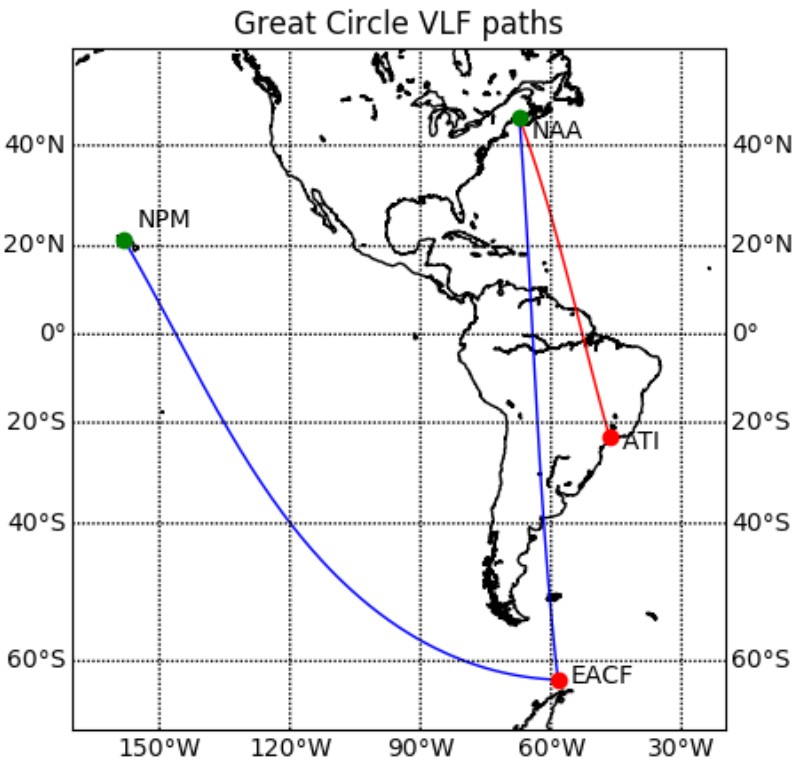

Figure 1: VLF propagation paths from NAA and NPM transmitters to the receiver stations located at Comandante Ferraz Brazilian Antarctic Station (EACF) (blue paths) and Atibaia, São Paulo (red path).

The wavelet analysis was applied to the VLF data obtained at EACF during the night of June 10, 2007, when a GW event was observed with a co-located airglow imager. This was done to compare the wave period and event duration parameters obtained from VLF data with the ones obtained from all-sky images.

Figure 2 shows two processed airglow images over plotted in geographical coordinates, centered at Comandante Ferraz Station (denoted by the red symbol), and observed on the night of 09-10 July 2007 when it was possible to identify a gravity wave event (inside the white box) in the upper mesosphere by

using a wideband near-infrared hydroxyl (OH-NIR) filter. The wave propagation direction is denoted by the arrow put just ahead of the box in the first image. The date and time of observation are indicated in the top of the map, the latitude and longitude (apart at each 2 degrees) are also shown, as well as the horizontal distances (in km), respectively in latitude and longitude, just above and on the left of the airglow images, for distances of 2 degrees (in lat.) and 4 degrees (in lon.). The images were processed as follows: star's field subtraction, correcting for the fish-eye lens format, and application of the Time Difference (TD) image processing to a short set of images. The small projected area (312X312 km, resolution of 1 km/pixel) was caused by the limitations of the CCD's size relative to the optical system since this is a low-cost CCD that was adapted in an old optical system (nowadays the optical system was reassembled to allow a useful area in the CCD of 512X512 pixels).This mesospheric GW was classified by Bageston et al. (2011) as a mesospheric front observed at EACF from about 23:20 LT (LT=UT-3) up to 23:53 LT. The analysis was performed from 23:20 to 23:42 LT when it was visible an increase in the number of wave crests in the wave packet when it propagates across the field-of-view of the sky, and this growth rate was inferred as 4 waves crest per hour (Bageston et al., 2011). The FFT-2D spectral analysis was applied to six images from 23:32 to 23:38 LT on 9 July (2:32 to 2:38 UT on 10 July), and it was obtained the following wave parameters: horizontal wavelength of 33 km, observed period of 6 min and observed phase speed of 92 ms$^{-1}$. During the same night this event was observed with a co-located near zenithal (field of view about 22° off-zenith) temperature airglow imaging spectrometer, which observes the OH (6-2) band emission (FotAntar-3, Bageston et al., 2007). The spectral analysis of the temperature showed evidence of gravity waves of small scale with predominant period of ~14 min (Bageston et al., 2011). Since the spectrometer has a smaller field-of-view (~70 km in diameter) compared to the all-sky imager (~300 km of diameter in the un-warped images), the larger predominant periodicity obtained from the temperature could be one component of the main wave observed with airglow all-sky imager (Bageston et al., 2011). These parameters are similar to the ones obtained for mesospheric fronts or bore-type events, which were understood as a rare type of gravity waves at polar latitudes and was first observed at Halley Station on May 2001 (Nielsen et al., 2006).Nowadays, with more observations, it is clear that the mesospheric fronts or bores are more likely to be observed from middle to high latitude (even at unexpected places as the South Pole) as can be noted in the recent studies on this subject (e.g.: Pautet et al., 2018a, Giongo et al, 2018,Hozumi et al., 2018).

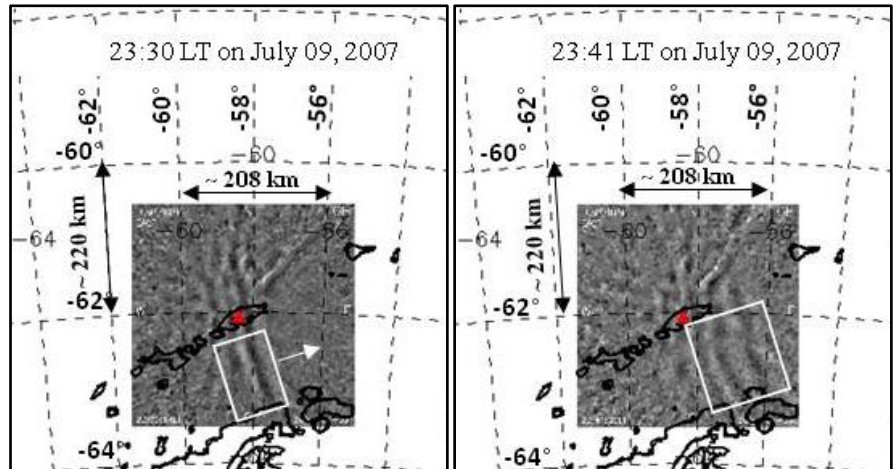

Figure 2: Processed all-sky images of the GW event observed at EACF (red symbol) at 23:30 and 23:41 LT (UT-3) on the night of 9-10 July 2007, showing the mesospheric front (white box) propagating from west/southwest to east/northeast (arrow direction in the first image). The images were projected at the mesospheric layer in order to have a spatial area as good as 312 X 312 km without significant distortion in the unwarped images.

The VLF amplitude from NAA transmitter detected at EACF on July 10, 2007 is shown in Fig. 3, where the vertical lines identify the sunrise and sunset hours at the transmitter (SR-T and SS-T, full lines) and receiver (SR-R and SS-R, dashed lines) stations. The wavelet spectral analysis (Fig. 4) was applied to the VLF data from 01:00 to 04:30 UT (22:00 LT June 9 to 01:30 LT on June 10, box in Fig. 3), which covers the nighttime interval of the images obtained with the co-located all-sky imager.

Figure 4 shows the spectral analysis applied to the VLF amplitude data. The analysis is applied to the residual value obtained after subtracting the raw data from a 12-min running mean (Fig. 4a), which implies in an upper cut-off period of ~30 min in order to characterize the small-scale and short period waves. Figure 4a shows clearly 4 strong fluctuations in the VLF amplitude between 1:50 and 2:40 UT (22:50 and 23:40 LT), which occurred in close temporal association with the crests identified in the airglow images. The last VLF fluctuation was the strongest one and has ended at ~02:40 UT (23:40 LT), near the time when the wave packet started to dissipate as observed in the airglow images (Bageston et al., 2011). The power spectrum of the residual VLF amplitude (Fig. 4b) shows strong significant components with periods between 4 and 16 min, with stronger peaks at ~6 min and 14 min. The global wavelet spectrum (Fig. 4c) shows a stronger component with period between 4 and 8 min that is due six significant events of ~20 min duration (Fig. 4d), with one of them occurring from 2:32 to 2:38 UT (23:32 to 23:38 LT) that is the same time interval a wave period of 6 min was identified in the airglow images. The other significant component with peak at ~14 min is present from 01:50 to 02:40 UT (Fig. 4d), the same time interval when the 4 crests of the mesospheric front were identified in the airglow images, and occurred in close temporal association with the identification of gravity waves with the same period in the

spectral analysis of the OH temperature obtained with the co-located imaging spectrometer (Bageston et al., 2011).

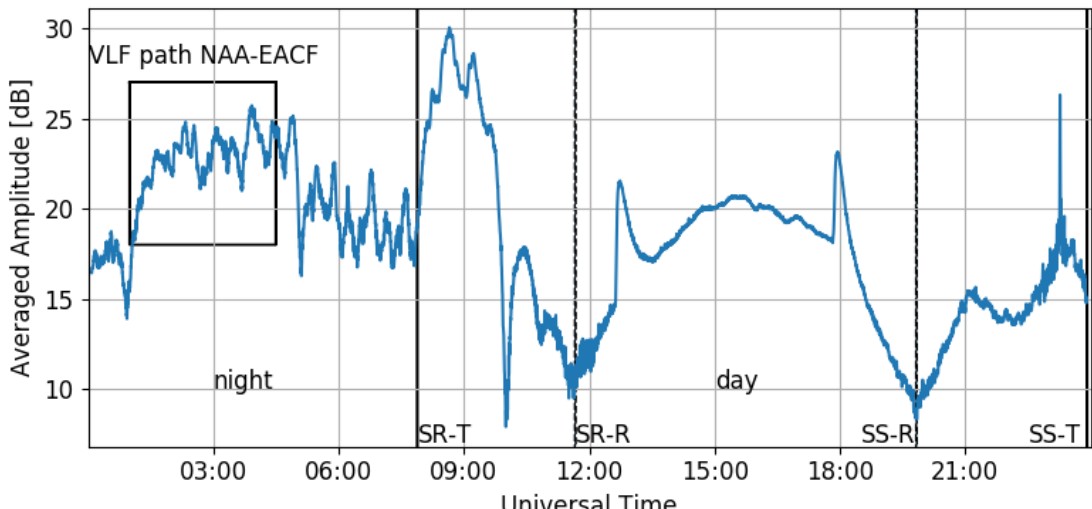

Figure 3: VLF amplitude from NAA transmitter station detected with 15 seconds time resolution at EACF on July 10, 2007. The vertical lines mark the sunrise (SR) and sunset (SS) at NAA transmitter station (T, full lines) and at receiver station (R, dashed lines). The periods of completely night and day in the NAA-EACF VLF path are identified. The box marks the time interval of data used to perform the spectral analysis.

Since the VLF path is quite long, we have performed one test to mak sure that the wave event was the same detected near EACF and not at any other location in the path between the transmitter and receiver. This test considers the wavelet analysis applied to the VLF path NAA-Atibaia (NAA-ATI), which is almost the same trajectory of NAA-EACF but its length is ~50 % shorter. Figure 5 shows no wave events at the time the event detected in the NAA-EACF path, that had association with the GW seen in the airglow imager, evidencing that the event occurred in the part of the VLF trajectory closer the EACF station. This test confirms the GW events detected by VLF technique in the NAA-EACF path occurred near Antarctic Peninsula and could be associated with the events observed by the airglow imager operating at EACF.

The characterization of the GWs using VLF amplitude data using wavelet analysis demonstrated the viability for the usage of VLF signals to obtain the period and time duration of the GW events. The use of VLF observations to characterize the GW events permit to obtain their climatology all over years since they do not are affected by the atmospheric conditions and also can be done during daytime.

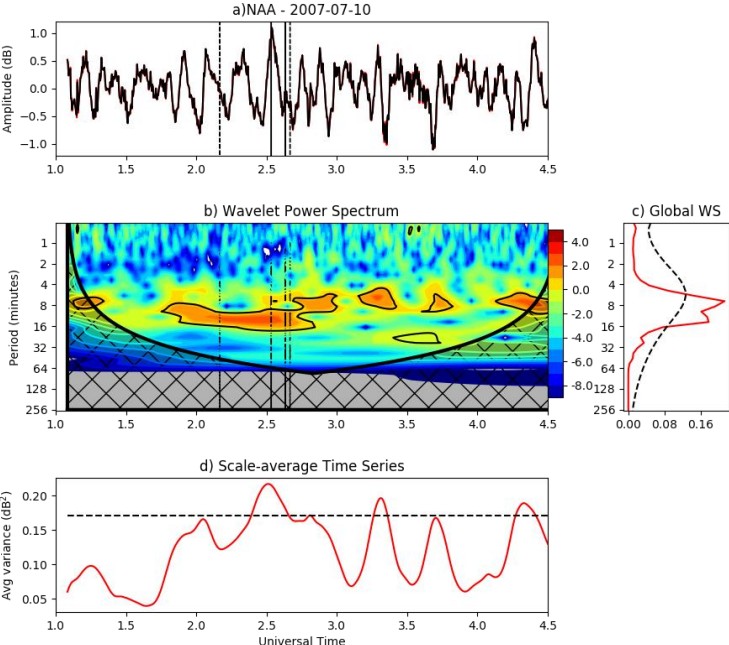

Figure 4: Example of wavelet spectral analysis applied to the VLF amplitude signal in the NAA-EACF GCP on 10 June 2007. (a) The residual VLF amplitude after subtracting the raw data from a10-min running mean. (b) Wavelet power spectra in logarithm (base 2), with regions of confidence levels greater than 95 % (showed with black contours), and the cross-hatched areas indicating the regions where edge effects become important. (c) time-averaged wavelet power spectra (Global WS). (d) scale-averaged wavelet power.

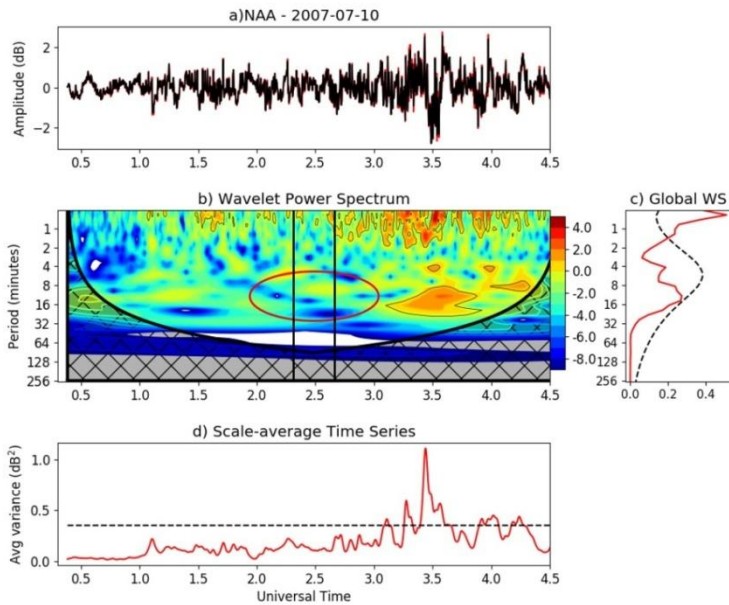

Figure 5: Same as Fig. 4, but for the VLF signal propagating in the NAA-Atibaia VLF GCP.

### 2.2 Climatology of GW period from VLF signal

The GW climatology was made based on the wavelet analysis applied to the VLF amplitude signal detected during both the nighttime and daytime hours in the NAA-EACF GCP for the full 2007 year. The wave period from VLF technique is the predominant component with the highest relative power amplitude in the global wavelet power spectrum. For example, in the analysis done in the previous sub-section, the predominant wave period was ~6 min. The wave period year-round climatology obtained via VLF technique during nighttime is compared with the one obtained with the co-located airglow imager.

### 3 Observational results

Here it is presented the statistical analysis of the predominant wave period of the GW events detected in the low ionosphere as amplitude fluctuations of the VLF signals, which is a new aspect of using the VLF technique. The analysis uses the VLF signal received at EACF during all-year around of 2007, and is performed independently for night (21:00 – 5:00 LT) and day (11:00 – 16:00 LT) hours, in order to avoid the influence of the sunrise and sunset terminators in the spectral analysis. The nighttime wave period properties obtained via VLF was used to compare with the wave period characteristics obtained with the co-located airglow all-sky imager.

The solar activity during 2007 was at lower levels since this year was near the minimum phase of the 23[rd] solar cycle. So it was a period of low occurrence of solar flares, and mostly of them of GOES C-class. In order to avoid the effect of D-region electron density changes associated with flares, the periods disturbed by the impact of flares were not used in the daytime wavelet analysis of VLF signal. The geomagnetic conditions were at lower levels during 2007 with 85% of the geomagnetic storms having the Dst index peak higher than -50 nT (weak storm) and only two moderate storms with Dst peak ~ -70 nT. The monthly Dst values were higher than -15 nT and kp lower than 2, which means low level geomagnetic activity.

Figure 6 shows the seasonal variation of GWs occurrence rate per month evaluated from the number of VLF observed days per month (black bars) and the respective number of nights (Figure 6a) and days (Figure 6b) with events detected in the low ionosphere. Small-scale GW events were detected at all nights and days of observations, with the predominant wave periods between 0 and 25 min, which are distributed in 5 period ranges from 0 to 25 min (0-5, 5-10, 10-15, 15-20 and 20-25). The occurrence rate of the events detected during nighttime (Figure 6a) shows the waves with period between 5 and 10 min occurred in higher number from May to September (>60%, winter season), followed by the waves with period between 10 and 15 min that suggest an equinoctial distribution occurring in higher number from October-November, and by the waves with periods < 5 min (AGWs) that also suggest an equinoctial distribution but with higher occurrence on March. The distribution of the waves with period between 15 and 25 min suggests a higher occurrence from October to March (Antarctic summer season). The distribution of the GWs with periods from 5 to 10 min is in an excellent agreement with the statistical results of the GW events observed by the co-located airglow all-sky imager, which showed the majority of the waves (~85 %) were observed between June and September (Bageston et al., 2009). The daytime

analysis (Figure 6b) shows the AGWs (period < 5 min) predominate all days (100%) from November to April, while the waves with period between 5 and 10 min were predominant in some days from May to October with higher occurrence between June and July (Antarctic winter season), followed by the waves with period between 10 and 20 min dominates for only a few days from May to August with lower occurrence in July.

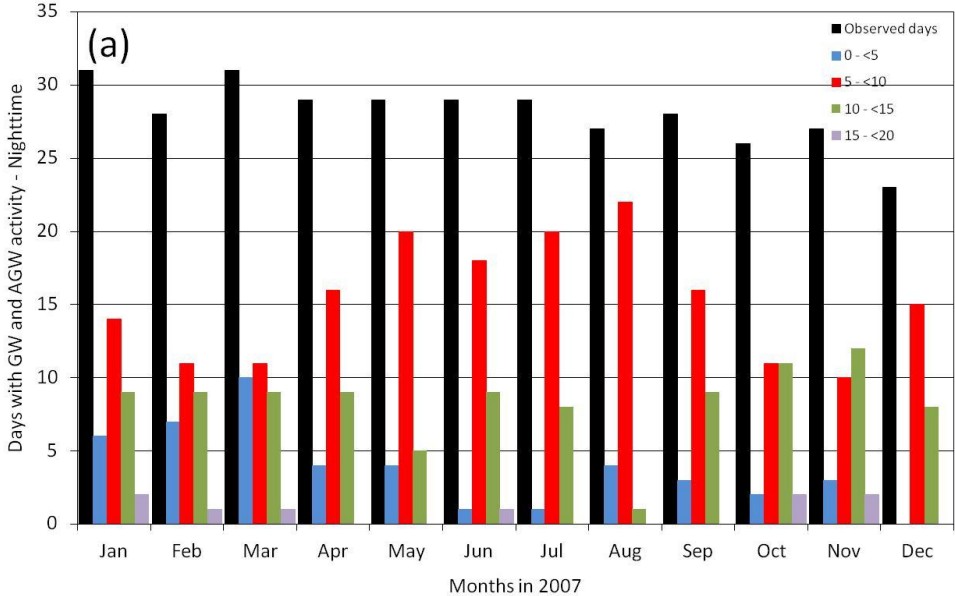

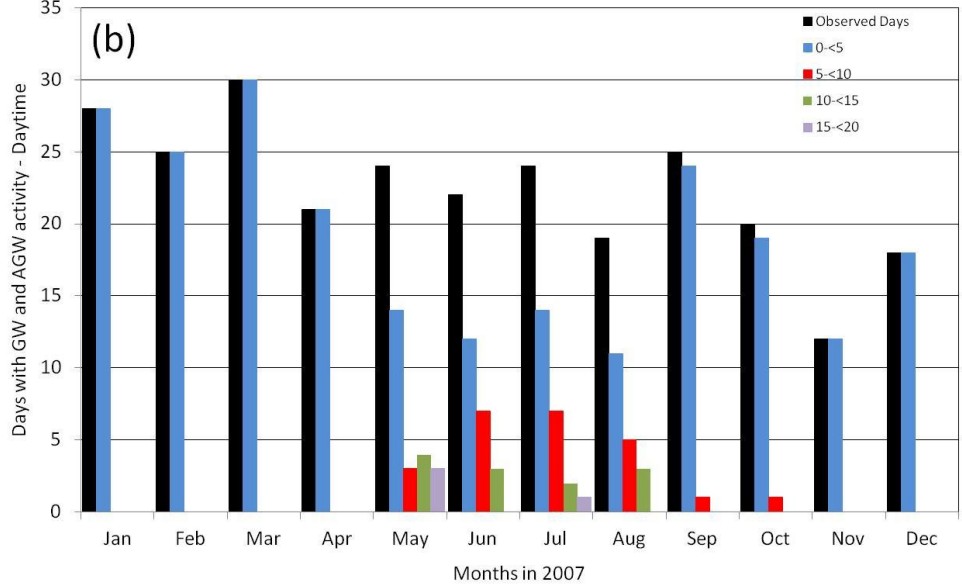

Figure 6: Monthly small-scale wave activity at EACF as detected in the low ionosphere using VLF technique during 2007 during night (a) and daytime (b) hours. The black bar shows the number of the observed days per month, and the colored bars show the number of nights and days per month with AGW events observed according to the wave predominant periods. The bar colors give the number of waves with predominant period observed in each month separated as the following period intervals: 0-5 min (blue), 5-10 min (red), 10-15 min (green), 15-20 min (purple).

Figure 7 shows the histogram plots containing the distribution of the predominant wave period of the wave events detected in the lower ionosphere using the VLF technique for the 337 nights and 268 days of observations in 2007. During nighttime (Figure 7a), the predominant wave periods were mostly distributed between 5 and 15 min (~ 80 %), with a higher number of occurrences between 5 and 10 min (~50 %), and smaller occurrence (~10%) of waves with periods below 5 min (AGWs). This wave period distribution for small-scale and short-period GWs is in good agreement with the statistics reported by Bageston et al. (2009) from the analysis of 234 GWs observed with a co-located airglow all-sky imager from April to October 2007. On the other hand, during the daytime the predominant wave periods were concentrated between 0 and 5 min (~85 %), in the AGWs range, followed by the GWs with period between 5 and 10 min (~10 %), and few waves (~5 %) with periods between 10 and 20 min.

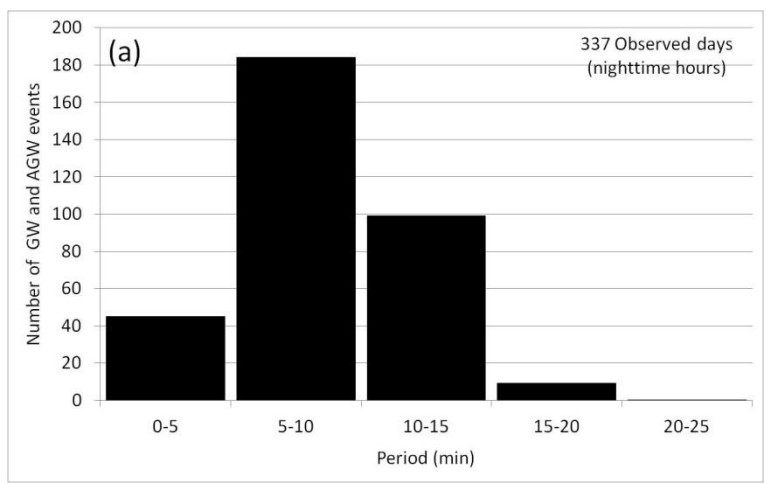

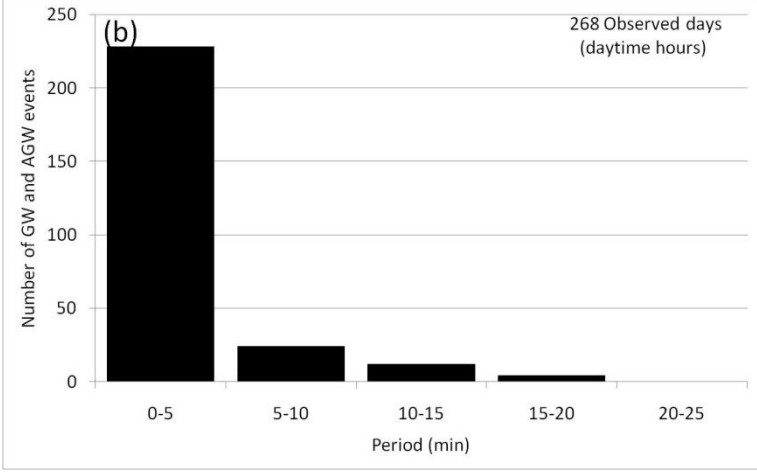

Figure 7: Histogram plots of the predominant observed wave periods of the small-scale AGWs detected in the lower portion of the ionosphere as amplitude variations of the VLF signal propagating in the NAA-EACF during night (a) and daytime (b) hours.

## 4 Summary

In this work we presented an investigation of the GWs characteristics in the low ionosphere, where they produce density fluctuations that were detected as amplitude variations of VLF signals. The analysis used the VLF signal transmitted from the US Cutler/Marine (NAA) station that was received at Comandante Ferraz Brazilian Antarctic Station (EACF), with is a great circle path crossing longitudinally the Drake Passage. The wavelet analysis of the VLF amplitude considered the predominant small-scale wave periods observed during the daytime and night hours separately, in order to compare the wave periods observed during nighttime with the ones obtained from a co-located airglow all-sky imager. The use of the VLF technique was validated by comparing the wave period and duration properties of one GW event observed simultaneously with a co-located airglow all-sky imager.

The statistical analysis of the wave period of the GW events detected at EACF using VLF technique for 2007 showed that the GW events were observed almost all days with VLF observations. During nighttime the waves with periods between 5 and 10 min are dominant (55 %) presenting a higher occurrence rate (large activity) per month from May to September with the maximum in June-July. The next predominant more frequent waves have periods ranging from 10 to 15 min (30 %) followed by few events (10 %) with periods lower than 5 min (AGWs), these both waves suggested an equinoctial distribution with the waves with periods between 10 and 15 min occurring in higher number in November and the shorter-period waves in March. The wave period distribution of the 5-10 min component is in good agreement with the wave period distribution of the GW events observed during 2007 with the co-located airglow all-sky imager. On the other hand, during daytime the waves with period below 5 min are dominant (85 %), and particularly from November to April they dominated all days of the months; followed by the waves with period between 5 and 10 min (10 %), which dominate for a few days from May to October and presents a higher occurrence from June to July, and finally for the waves with periods between 10 and 20 min (5 %) that dominates just for fewer days from May to August with lower occurrence rate in July.

These results show that VLF technique is a powerful tool to obtain the wave period and duration of GW events in the low ionosphere, with the advantage to be independent of sky conditions, and can be used during all day and year-round. The VLF technique also shows its potentiality to simultaneously obtain the properties of the AGWs and GWs, which is important to better define the generation mechanisms of these atmospheric waves and their relevance in the Earth's thermosphere. The analysis of wave events using VLF signals from two distinct transmitter stations apart ~100$^o$ in longitude, for example NAA-EACF and NPM-EACF could also be used to obtain information about the velocity and direction of propagation of the GW events, but these tasks will be treated in future works.

*Competing interests.* The authors declare that they have no conflict of interest.

*Acknowledgments.* EC thanks the National Council for Scientific and Technological Development - CNPq (processes no: 556872/2009-6, 406690/2013-8 and 306142/2013-9), São Paulo Research Foundation – FAPESP (process no: 2019/05455-2) for individual research support, and the National Institute for Space

Research (INPE/MCTI). The authors also acknowledge the support of the Brazilian Ministries of Science, Technology, Innovation and Communications (MCTIc), Environment (MMA) and Inter-Ministry Commission for Sea Resources (CIRM). LTMR thanks the Coordenação de Aperfeiçoamento de Pessoal de Nível Superior - Brasil (CAPES) - Finance Code 001.

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

Figure Captions

Figure 1: VLF propagation paths from NAA and NPM transmitters to the receiver stations located at Comandante Ferraz Brazilian Antarctic Station (EACF) (blue paths) and Atibaia, São Paulo (red path).

Figure 2: Processed all-sky images of the GW event observed at EACF (red symbol) at 23:30 and 23:41 LT (UT-3) on the night of 9-10 July 2007, showing the mesospheric front (white box) propagating from west/southwest to east/northeast (arrow direction in the first image). The images were projected at the mesospheric layer in order to have a spatial area as good as 312 X 312 km without significant distortion in the unwarped images.

Figure 3: VLF amplitude from NAA transmitter station detected with 15 seconds time resolution at EACF on July 10, 2007. The vertical lines mark the sunrise (SR) and sunset (SS) at NAA transmitter station (T, full lines) and at receiver station (R, dashed lines). The periods of completely night and day in the NAA-EACF VLF path are identified. The box marks the time interval of data used to perform the spectral analysis.

Figure 4: Example of wavelet spectral analysis applied to the VLF amplitude signal in the NAA-EACF GCP on 10 June 2007. (a) The residual VLF amplitude after subtracting the raw data from a10-min running mean. (b) Wavelet power spectra in logarithm (base 2), with regions of confidence levels greater than 95% (showed with black contours), and the cross-hatched areas indicating the regions where edge effects become important. (c) time-averaged wavelet power spectra (Global WS). (d) scale-averaged wavelet power.

Figure 5: Same as Figure 4, but for the VLF signal propagating in the NAA-Atibaia VLF GCP.

Figure 6: Monthly small-scale wave activity at EACF as detected in the low ionosphere using VLF technique during 2007 during night (a) and daytime (b) hours. The black bar shows the number of the observed days per month, and the colored bars show the number of nights and days per month with AGW events observed according to the wave predominant periods. The bar colors give the number of waves with predominant period observed in each month separated as the following period intervals:  0-5 min (blue), 5-10 min (red), 10-15 min (green), 15-20 min (purple).

Figure 7: Histogram plots of the predominant observed wave periods of the small-scale AGWs detected in the lower portion of the ionosphere as amplitude variations of the VLF signal propagating in the NAA-EACF during night (a) and daytime (b) hours.