# Peer review of "Characterization of gravity waves in the lower ionosphere using VLF observations at Comandante Ferraz Brazilian Antarctic Station"

_Annales Geophysicae, 2019_

## Referee Comment (RC1) · Anonymous Referee #1 · 9 Oct 2019

The present work demonstrates usefulness of VLF technique to observe GW activity in the lower ionosphere ($\sim$70 km ?), using VLF receiver located at EACF, Antarctica. For validation they compared the results with airglow image data from the MLT region ($\sim$90 km). They could get a large amount of periodic oscillation events in 2007, and could investigate seasonal variation of GW events. The VLF technique itself is not new, but it is a new aspect to use it for monitoring GW activity. The authors successfully could do it. The figure 6 (seasonal variation of GW occurrence) is worth to discuss as a new result. I think the present manuscript is worth to publish in Journal. However some minor revisions would be necessary before to be accepted, as listed below:

Page 2, line 14, higher than: longer than ? Page 4, Line 21-22: The authors mention that simultaneous measurement of VLF over NAA-EACF and NPM-EACF gives opportunity to identify propagation direction of GW. However they did not use it in this work. Why ? Page 6, Figure 2: I think it is better to show only the lower panel (geographically coordinated images), with the latitude and longitude scales, so that readers can understand in the horizontal coverage of the images. Page 10, Figure 6, the authors presented GW occurrence rate using only nighttime VLF amplitude oscillation. They did not mention about how was the occurrence during the day time. Discussion of the data including the daytime occurrence would be worth. Page 11, line 23-25: The simultaneous analysis of VLF,,,: The authors did not show any analysis in the manuscript. (END of Review)

---

## Author Comment (AC1) · 29 Oct 2019

The response to the Referee#1 (Answer_Referee#1), the manuscript with the changes identified (ECorreia_2019-ANGEO-REC1-Commented-Version) and the revised manuscript (ECorreia_2019-ANGEO-REC1-RevisedVersion) are in the Supplement.zip file.

Please also note the supplement to this comment:
https://www.ann-geophys-discuss.net/angeo-2019-123/angeo-2019-123-AC1-supplement.zip

---

## Referee Comment (RC2) · Anonymous Referee #2 · 14 Nov 2019

Comment to Authors on manuscript: "Characterization of gravity waves in the lower ionosphere using VLF observations at Comandante Ferraz Brazilian Antarctic Station" by Emilia Correia et al.

Review: This is a nice piece of work which demonstrates usefulness of VLF signal in understanding the gravity waves contribution in dynamics of lower part especially D-region of the ionosphere. As a cross verifications authors also use airglow observations and this a new content in the manuscript.

In my review I followed and read the revised version of the manuscript submitted by Emilia Correia et al. The comments provided by Reviewer 1 were almost similar to my

comments when I read the manuscript before revision.

Hence I do not give any major comments except one comment: Authors are totally silent in the manuscript on the geomagnetic condition from solar origin. Please note that activities like solar flares even of the modest C-class are known to change the D-region electron density concentration especially during daytime. Did authors considered this aspect in their analysis? It will be of significance to include the geomagnetic condition during the days of analysis presented in manuscript.

The minor comments is listed as below: [Authors to follow their revised manuscript for corrections as listed] 1. Page 2, Line 7: which is a great circle path -> with its great circle path

2. Page 2, Line 27: In the last decades -> During last decades

3. Page 4, Line 18: Earth-ground cavity -> via multiple reflections

4. Page 5, Line 4: It is used the tool developed by Torrence and Compo (1998) and including the rectification -> The tool used is developed by Torrence and Compo (1998) and includes the rectification

5. Page 13, Line 1: which is a great circle path -> with its great circle path

Final Comments: The manuscript can be accepted after addressing the comments listed above

Please also note the supplement to this comment:
https://www.ann-geophys-discuss.net/angeo-2019-123/angeo-2019-123-RC2-supplement.pdf

---

## Author Comment (AC5) · 19 Dec 2019

The revised manuscript considering the comments and suggestions of both referees is already concluded and submitted for evaluation.

The files with the point-to-point answers and the marked version, as well the revised version were already uploaded.

---

## Author Response (AR1)

Authors Response

Point-to-point response to the reviews and a marked version identifying the changes in the manuscript.

**Point-to-point responses**

Answer to Referee#1

We would like to thank Referee#1 for the very fruitful comments and suggestions. In the following we include our answers point-by-point.

The new texts are in green and replace the old ones marked as strikethrough in red.

Referee Comment: 'The VLF technique itself is not new, but it is a new aspect to use it for monitoring GW activity.'

*This comment was considered in Page10, lines 3-5*

Minor revisions:

1. Page 2, line 14, higher than: longer than?

*All the abstract was rewritten face the revisions done in the manuscript, now considering the GWs daytime occurrences and also the inclusion of waves with periods below 5 min (AGWs).*

2. Page 4, Line 21-22: The authors mention that simultaneous measurement of VLF over NAA-EACF and NPM-EACF gives opportunity to identify propagation direction of GW. However they did not use it in this work. Why ?

*This mention in the text was moved to conclusions because is another interesting information possible to be obtained but not considered in this work. It deserves more attention to identify clearly the waves we want to follow since in the VLF signal we observe various waves simultaneously. See Pag 14, lines 17-19 in the commented version.*

3. Page 6, Figure 2: I think it is better to show only the lower panel (geographically coordinated images), with the latitude and longitude scales, so that readers can understand in the horizontal coverage of the images.

*Figure 2 was changed accordingly.*

4. Page 10, Figure 6, the authors presented GW occurrence rate using only nighttime VLF amplitude oscillation. They did not mention about how was the occurrence during the day time. Discussion of the data including the daytime occurrence would be worth.

*Following the referee suggestion, the statistical analysis now also includes the wave occurrences during daytime hours. The text was rewritten including now the wave occurrences separately in night and daytime, and also considering waves with periods below 5 min, which are associated with the acoustic waves. See in the commented version page 2, lines 26-30. Page 3, lines 8-9 and lines 37-39. Page 10, lines 14-16 and 36-38. Page 11, lines 2-7 and from 21-39. Page 12, lines 20-26. Page 13, lines 7-16, and all Summary.*

5. Page 11, line 23-25: The simultaneous analysis of VLF„„: The authors did not show any analysis in the manuscript.

*Already commented above. See Page 14, lines 17-19 in the commented version.*

Answer to Referee#2

We would like to thank Referee#2 for the comments and suggestions. In the following we include our answers point-by-point.

The new texts are in green and replace the old ones marked as strikethrough.

Referee Comment: Hence I do not give any major comments except one comment: Authors are totally silent in the manuscript on the geomagnetic condition from solar origin. Please note that activities like solar flares even of the modest C-class are known to change the D-region electron density concentration especially during daytime. Did authors considered this aspect in their analysis? It will be of significance to include the geomagnetic condition during the days of analysis presented in manuscript.

*This comment was considered in Page10, lines 17-24*

Minor comments:

1. Page 2, Line 7: which is a great circle path -> with its great circle path

*Done*

2. Page 2, Line 27: In the last decades -> During last decades

*Done*

3. Page 4, Line 18: Earth-ground cavity -> via multiple reflections.

*Done*

4. Page 5, Line 4: It is used the tool developed by Torrence and Compo (1998) and including the rectification -> The tool used is developed by Torrence and Compo (1998) and includes the rectification

*Done*

5. Page 13, Line 1: which is a great circle path -> with its great circle path

*Done (Page 13, line 5)*

Marked_Version with the changes done accordingly the referees comments and suggestions

[revised manuscript text omitted]